# Nitric Oxide Reverses the Position of the Heart during Embryonic Development

**DOI:** 10.3390/ijms20051157

**Published:** 2019-03-07

**Authors:** Jamila H Siamwala, Pavitra Kumar, Vimal Veeriah, Ajit Muley, Saranya Rajendran, Salini Konikkat, Syamantak Majumder, Krishna Priya Mani, Suvro Chatterjee

**Affiliations:** 1Vascular Biology Lab, AU-KBC Research Centre, Anna University, Chennai, Chennai-600044, India; jamila_siamwala@brown.edu (J.H.S.); jadaunpavitra@gmail.com (P.K.); vimalgene@gmail.com (V.V.); aamuley@gmail.com (A.M.); saranr77@gmail.com (S.R.); salinikrish@gmail.com (S.K.); sam_sya2004@yahoo.co.in (S.M.); krishnabio86@gmail.com (K.P.M.); 2Department of Molecular Pharmacology, Physiology and Biotechnology, Brown University, Providence, RI 02903, USA; 3Vascular Research Laboratory, Providence Veterans Affairs Medical Center, Providence, RI 02908, USA; 4Department of Biotechnology, Anna University, Chennai, Chennai-600025, India

**Keywords:** nitric oxide, situs inversus, BMP4, CPC migration

## Abstract

Nitric oxide (NO) produced by endothelial nitric oxide synthase (eNOS) plays crucial roles in cardiac homeostasis. Adult cardiomyocyte specific overexpression of eNOS confers protection against myocardial-reperfusion injury. However, the global effects of NO overexpression in developing cardiovascular system is still unclear. We hypothesized that nitric oxide overexpression affects the early migration of cardiac progenitor cells, vasculogenesis and function in a chick embryo. Vehicle or nitric oxide donor DEAN (500 μM) were loaded exogenously through a small window on the broad side of freshly laid egg and embryonic development tracked by live video-microscopy. At Hamburg Hamilton (HH) stage 8, the cardiac progenitor cells (CPC) were isolated and cell migration analysed by Boyden Chamber. The vascular bed structure and heart beats were compared between vehicle and DEAN treated embryos. Finally, expression of developmental markers such as BMP4, Shh, Pitx2, Noggin were measured using reverse transcriptase PCR and in-situ hybridization. The results unexpectedly showed that exogenous addition of pharmacological NO between HH stage 7–8 resulted in embryos with *situs inversus* in 28 out of 100 embryos tested. Embryos treated with NO inhibitor cPTIO did not have *situs inversus*, however 10 embryos treated with L-arginine showed a *situs inversus* phenotype. N-acetyl cysteine addition in the presence of NO failed to rescue *situs inversus* phenotype. The heart beat is normal (120 beats/min) although the vascular bed pattern is altered. Migration of CPCs in DEAN treated embryos is reduced by 60% compared to vehicle. BMP4 protein expression increases on the left side of the embryo compared to vehicle control. The data suggests that the NO levels in the yolk are important in turning of the heart during embryonic development. High levels of NO may lead to *situs inversus* condition in avian embryo by impairing cardiac progenitor cell migration through the NO-BMP4-cGMP axis.

## 1. Introduction

Furgott & Zawadski (1980)’s discovery of the physiological roles of nitric oxide (NO) gas challenged scientific conventions and a multitude of processes influenced by NO gas were brought to light [1]. NO, an endothelial cell derived signalling molecule is synthesized by L-arginine and produced by conserved nitric oxide synthases: endothelial NOS (eNOS), inducible NOS (iNOS), neuronal NOS (nNOS) and mitochondrial NOS (mtNOS) [2,3,4]. Optimal NO signalling is vital for cardiovascular homeostasis, regulation of vascular tone [5,6,7], platelet aggregation [8], inhibition of vascular smooth muscle cell proliferation [9] and protection against ischemia reperfusion injury. Reduction in NO levels through NO degradation, reduced NO synthesis or desensitization of NO are associated with cardiovascular diseases. Conversely, strategies aimed at improving NO bioavailability and signalling of NO confer cardioprotection [10]. Inhaled NO is used for the treatment of pulmonary hypertension in new born, pre-term neonates with hypoxic respiratory failure [11] and phosphodiesterase inhibitors (PDE) for example, sildenafil therapy to improve hemodynamics in the children with pulmonary hypertension [12]. However, therapeutic modulation of NO levels has limited success as the NO action depends on dose, time and location of delivery. The consequences of NO overdosing in treatment strategies are largely unknown. Also, the effects on the cardiovascular development of the foetus exposed to high levels of NO due to infections induced inflammatory responses and NO production are unclear.

As we were developing novel techniques to utilize the fertilized egg as a model of angiogenesis and drug screening, we discovered serendipitously, that the eggs treated with NO donors showed heart situated on the right side instead of the left side of the embryo. Therefore, we hypothesized that NO influences the establishment of the left-right position of the heart and vessels through cardiac progenitor cells migration. 

The mechanisms of NO signalling are exhaustive and influence several biological processes. Canonically, NO activates soluble guanyl cyclase (sGC) downstream and catalyses the synthesis of cGMP, a second messenger which targets a number of proteins including bone morphogenetic protein-4 (BMP4) involved in determining the positioning of the heart during embryonic cardiovascular development. BMP4 is involved in migration of cardiac progenitor cells (CPC) on the left side of the embryo [13]. In zebra fish, CPC, actively migrate to cause leftward extension of the cardiac tube in a process referred to as cardiac jogging. Prior to cardiac jogging the BMPs are strongly expressed within the heart cone [13]. BMP represses CPC migration on the left resulting in an increase in the CPC migration on the right side and the formation of the heart tube on that side. To clarify the role of NO in the BMP4 mediated CPC migration and the positioning of the heart, we hypothesized that NO overproduction affects the early migration of cardiac progenitor cells and positioning of the heart through BMP4 mediated mechanism in a chick embryo.

## 2. Results

### 2.1. Situs Inversus Occurs with Nitric Oxide Overexpression in Chick Embryos

NO, another gaseous molecule was able to cause *situs inversus* in about 20–30% of the chick embryos although the mode of delivery of NO was different from Fujinaga’s work (Figure 1A). PBS or NO donors were administered in the same way as commonly used by the pharmacist to generate vaccine by applying the viral load through a small hole on the broad end of a fertilized egg in close proximity to the embryo. Treatment with 500 µM DEAN having a half-life of 2min exerted maximal effect on the situs of the embryos (Figure 1B) compared to other concentrations of DEAN (100 µM or 1000 µM) or another NO donor (SNP 500 µM). DEAN concentrations above 1000 µM resulted in lethality. Addition of DEAN between 0 to 24 h of incubation showed maximum number of orientations suggesting a critical window for the left to right orientation (Figure 1C). At the Hamburger-Hamilton (HH) stage 24 after the cardiac looping stage, the eggs were opened and scored for abnormalities. The heart tube loops from left to right in a C-shaped loop when viewed from the ventral side facing embryo. Surprisingly in NO treated embryos, 20% of the live embryos (40/200) observed at HH24 (4 days post incubation) had reversed heart looping (D shaped loop) whereas all the PBS treated embryos had normal heart looping (*n* = 200 embryos). As the looping of the heart determines the foetal position, all the PBS treated embryos curved in an anti-clockwise direction (C shape) (towards the right side of the embryo), while the NO treated embryos curved in a clockwise direction (D shape) (towards the left side of the embryo) (Figure 1A). The cardiac looping and turning of the chick embryo were followed live and bright field images of the heart rotation of live chick embryos were taken between 30–50 h of development (Figure 1E). Apart from the changes in cardiac looping, the main vitelline vessels developing from the right side of the embryo were now situated on the left side, thereby changing the overall blood circulation in the embryo. 

To confirm NO specific effects and to rule out the non-specific effects of DEAN, we used different NO donors. Between the NO donors with different half-lives tested (SNP 500 µM release in seconds, DEAN 500 µM release in 2min and DETA 500 µM release in 3 h), DEAN addition resulted in maximum number of *situs inversus* (SI) embryos (30/100eggs) followed by SNP (15/100 eggs) and DETA (1/100 eggs) (Figure 1B). When the exogenous NO was quenched using cPTIO, the embryos were rescued from NO mediated SI (Figure 1D). 

### 2.2. NO Causes Situs Inversus by Altering Cardiac Progenitor Cells Migration from the Blood Islands

Haematoxylin and eosin staining of vertical section of 4th day old embryonic heart shows rudiment interventricular septum and loss of ventricular polarity under NO treatment (Figure 2A). The cardiovascular development in a fertilized egg begins with the migration of cardiac progenitor from the blood islands a primary event in vasculogenesis and angiogenesis [14]. In our experiments, addition of NO at Hamburger-Hamilton (HH) stage 7–8 (24 to 28 h post incubation) resulted in the maximum number of embryos with *situs inversus* (Figure 1C,E), a time period corresponding to differentiation of cardiac progenitor cells (CPC) involved in blood island formation [14].

### 2.3. Arrhythmia and Abnormal Vascular Patterning and Function Occurs in Embryos with Situs Inversus

Next, heartbeats per minute were scored using a heartbeat monitor. NS embryos had higher heartbeats (tachycardia) than SI embryos (Table 1). NO treated embryos age faster compared to PBS treated embryos. No other morphological defects were observed in any other developing tissues or organ systems (Table 2). Other vascular parameters such as heart contraction area, heart relaxation area and vascular bed area were also recorded (Table 3).

### 2.4. NO Quencher cPTIO and BMP4 Inhibitor DM Restores Cell Migration

BMPs control CPC migration by pushing the cells to the right side of the embryonic axis. BMPs are strongly expressed in the heart (Figure 2B) prior to cardiac jogging however their expression is strictly controlled by Nodal. If the hypothesis that NO alters cellular movements at the node through Nodal repression and BMP4 activation is true, blocking BMP4 should resume cell migration. When the BMP4 signalling in the CPC is blocked using a BMP specific pharmacological inhibitor Dorsomorphin (DM), there is CPC migration and the cardiac field is pushed to the right side. Endogenous NO quenched using cPTIO, resulted in reduced cell migration as expected (Figure 2B). Similarly, when DM were added along with DEAN, the cell migration repression was removed. Further the CPC show NO mediated increases in hydrogen peroxide, peroxynitrite and superoxide blunted by the NO quencher cPTIO. Taken together, there is an overall increase in CPC cell migration and ROS production (Figure 2C), which pushes the cardiac field development on the left side instead of the right hence resulting in *situs inversus*.

### 2.5. BMP4 Inhibitors Rescue Zebrafish Embryos from Situs Inversus

Treatment of zebrafish embryo with DEAN followed by recombinant noggin protein, a selective inhibitor of BMP4, reduced the number of embryos with SI by 80% compare to that of DEAN (Figure 3F).

### 2.6. NO Causes Situs Inversus through BMP4-SMAD1 Mediated Pathway

DEAN treated embryos show a higher gene expression of BMP4 (Figure 3). Further in the embryos with a reverse heart position, BMP4 were expressed in the heart tissue of DEAN treated embryos but not in PBS treated embryos (Figure 3C). In-depth expression analysis of the genes involved in the left right patterning reveal modulation of BMP4 at HH6 (24 h) with an increase in expression in DEAN treated embryos. Shh is not expressed till stage HH6 (29 h) Pitx2, which controls BMP4 expression, shows a temporal increase in expression (Figure 3B). From HH11-HH15, BMP4 protein expression is expressed in heart tissue of right-handed side of SI embryos but not in the NS (9 out of the 24 embryos analysed) (Figure 3A). This data corresponds with the BMP4 mRNA expression observed in DEAN treated heart tissue in HH12 embryo (Figure 3B). Immunohistochemistry (IHC) of 4th day old embryonic heart confirms BMP4 antibody specificity. Noggin, a selective inhibitor of BMP4, (* *p* = 0.001) shows reduction in BMP4 expression indicating BMP4 antibody specificity (Figure 4A). Immunohistochemistry of 4th day chick embryonic heart further confirms higher expression of BMP4 protein in the presence of NO. The protein levels of BMP4 were significantly higher (* *p* < 0.05) under NO treatment (Figure 4B), whereas the expression of BMP4 antagonist; Noggin expression reduces upon DEAN treatment (Figure 3F). In order to test if BMP4 signalling is mediated through SMAD’s, SMAD phosphorylation profile were examined in heart sections. Immunostaining shows higher expression of pSMAD1 in the heart sections of DEAN treated embryos (Figure 4B).

## 3. Discussion

In this study, we show that NO levels during early development play a critical role in positioning of the heart on the right side of the developing embryo. High levels of NO switches the heart position from the right to the left resulting in *situs inversus*. NO reverses the cardiac development by altering the cardiac progenitor cell migration through phosphorylation of SMAD’s in a cGMP/BMP4 dependent manner.

eNOS knockout mice model show redistribution and vascularization of retina in the absence of optimal levels of NO levels [15] suggesting compensation and critical requirement of appropriate NO quantities for vascularization. The diffusible, gaseous nature of NO prompted us to use air sac model for the delivery of NO. We employed a system commonly used by the pharmacologist to treat diseases and to administer viral load to avian embryo in order to generate vaccines. NO donor drugs are thought to mimic the cellular effects of endogenously synthesized NO [16]. In our study we added NO exogenously by means of NO donor (DEAN) in the air sac at HH stages 4–6. The chick embryo is an excellent model for left-right determination studies, since the embryos can be manipulated as early as the blastula stage, independent of the maternal effects. Moreover, axis determination and organ primordial are conserved among mammals and birds.

The work of Lancaster (1994) show that NO can diffuse relatively long distance from the source [16]. NO levels were detected at the centre of the yolk close to the embryo after addition through the broad end of the egg. Among the different DEAN concentrations (100 to 1000 µM) used, 500 µM DEAN resulted in the maximum number of SI embryos (30%). DEAN concentration of 1000 µM resulting in lethality maybe due to accumulation of nitrosative stress and negative signalling. The final concentration of NO reaching embryo maybe lower because of free radical quenching property of the yolk. To rule out the non-specific effects of DEAN we used other NO donors having different half-lives and chemical structures. The results show that the orientations are not DEAN specific and half-life of NO donor is an important determinant in the process of the embryo heart looping. Chan (2004) et al. suggests that hypoxic exposure at critical window in development results in differential effects on organ development [17]. Preliminary studies showed that frequency of SI was higher in chick embryos (28%) than in zebrafish (10%) (Table 2) under DEAN treatment. Therefore, further studies were performed on chick embryo model. The maximum orientations seen in zebrafish embryo at 6hpf coincide with formation of the loop of the heart. This might explain why we see reduced number of orientations after 6hpf as the heart is already formed after this stage.

DEAN is added exogenously by means of NO donor but question remains as to the role of endogenous NO in a developing embryo under NO treatments? Exogenously added NO donors release NO either spontaneously or through enzymatic or non-enzymatic bioactivation in tissue similar to endothelium derived NO [18]. The endogenous levels of NO can be quenched by cPTIO, an NO scavenger and we see loss of function, that is, there were no situs inversus embryos. Endogenous NO is already shown to be responsible for early vasculogenesis, angiogenesis and maintenance of vascular homeostasis [19]. NO mediates its effects mainly through activation of soluble guanylate cyclase resulting in elevated cGMP. This is concomitant with the observation that if minimum concentrations of NO required activating guanylyl cyclase having equilibrium dissociation constant (*k*_dis_) equal to 0.25 µM fall significantly, then the biological function of NO is diminished. This in turn emphasizes the role of NO concentration in the development of an embryo. Exceeding the normal physiological levels of NO could perturb the signalling events leading to morphological anomalies.

The monocilia movement results in the leftward flow in the embryonic space leading to activation of nodal on the left-side of the paraxial mesoderm. Nodal in turn activates Pitx2 on the left-side. Thus, a coordination of the 4 major dimensions defining molecules such as right side, acting Snail specific molecules, left side, Nodal, Lefty 2 and Caronte, midline, Lefty 1 and laterally, BMP4 results in the establishment of the LR axis [20]. BMP4 is a downstream target of Pitx2 which is specifically implicated in heart looping during the late phases of LR determination [21,22]. The work of Monsoro Burq et al. (2001) show that BMP4 can simultaneously control the LR pathways thereby emphasizing the significance of this molecule in the establishment of the LR axis [23]. Interestingly patients of primary ciliary dyskinesia, which is further, associated with heterotaxy syndrome, shows abnormal NO profile in respiratory tract and atrioventricular nodal re-entrant tachycardia [24,25]. Therefore, we assume that controlled NO release during early phase of development either reduces the shear stress in amniotic cavity or directly acts on monocillia to modulate their movements. In the extension of the present work we would be probing NO implications in monocillia movements and how it defines left-right asymmetry. We also observe tachycardia in NO treated oriented embryos (Table 1) like the human conditions. Therefore, the present work offers an opportunity to establish a NO based model of primary ciliary dyskinesis associated *situs inversus*.

BMP4 and Sonic Hedgehog (Shh) is responsible for the asymmetric development of the chicken pro-epicardium (PE) and might reflect side-specific differences [26]. There are reports clearly showing that BMP4 activates iNOS expression in macrophages [27] and further NO acts upstream to BMP4 [21]. NO/cGMP pathway mediates estrogenic skeletal effects thereby playing a role in short term bone metabolism [28]. In the present study we found that NO promotes BMP4 expression in heart of chick embryos (Figure 3B–E). Increase in BMP4 signalling could alter the morphogenetic gradient that determines which signalling molecules will be expressed on the right side or left side of the embryo. BMPR IA and Smad1 are transiently expressed on the right side of Hensen’s node, when L-R polarity is being established [23]. Since NO activates BMP4, more BMP4 ligand will be available to bind to BMP4 receptors which in turn form hetero-oligomeric complexes with SMAD’s and translocate to the nucleus and initiate the transcription of left sided or right sided genes. There could also be a feedback regulation due to the increase in BMP4 expression, which in turn suppresses the expression of the right side and results in the left sided positioning of the heart and the embryo.

Shh is expressed on the left side of the node and is a master regulator. Any changes in Shh levels could have brought drastic changes in the morphology and positioning of organs as previously reported [29,30,31]. We however did not observe any change in the morphology and positioning of other organs under our experimental conditions (Table 2), consistent with the findings there was no change in Shh gene expression at the later stages of development (Figure 3B,D,E). Further on analysis of gene expression of BMP4 in different tissues such as the brain, heart, liver, eye and muscle of the 11th day embryo, we found that NO induced BMP4 gene expression in heart and muscle but not in other tissues (Figure 3C) substantiating earlier reports about the involvement of the position of the heart in determining the rotation of the embryo [32]. Shh and BMP4 transcription is negatively regulated by each other, resulting in complementarity between these two genes on either side of the node. Noggin is present in the midline antagonizes BMP4 [23] and prevents BMPs from binding to their receptors by binding tightly to BMPs and thereby inhibits BMP signalling [33,34]. Moreover, BMP4 is downstream to activin signals and controls Fgf8. Thus, early BMP4 signalling coordinates left and right pathways in Hensen’s node in the early phase of embryo development. However, at the 22nd stage, BMP4 expression becomes more asymmetric with far more on the left than on the right side of the heart tube in LPM [21]. In case of right-side expression of BMP4, a reversed jogging and looping of the heart tube has been reported [21]. In concordance with the earlier reports we found that noggin expression is low and BMP4 is over expressed on the right side of the oriented embryo and less expression was seen on the left side (Figure 5).

Recently it has been shown that BMP signalling exerts an anti-motility effect on cardiac progenitor cells (CPC) via the regulation of Non-muscle Myosin II (NHMII) on the left side [13]. We predict that NO interferes with the Nodal repression of the BMP on the left side. Hence there is an increase in CPC migration on left side and the entire cardiac field develops on this side instead of the right one. Preliminary data suggests that NO increases BMP4 expression in the heart (Figure 3B–E) whereas Nodal expression is repressed (Figure 3F). Nodal repression may remove the inhibition on BMP4 expression which in turn drives CPC motility as seen in Figure 5. Interestingly Wnt3a levels were also elevated which indicates that apart from CPC motility being elevated there is also an increase in cell proliferation as canonical wnt signalling controls the proliferation and differentiation of embryonic cardiac precursors into mature cardiomyocytes [35]. NO may influence BMP signalling through the generation of oxidative stress by reactive nitrogen species (RNS) and reactive oxygen species (ROS) comprising of hydrogen peroxide, peroxynitrite and superoxide. In summary, the NO mediated bending of primitive streak is attributed to faster cell movements on the right-hand side. When NO levels are increased by exogenously adding NO, there is an increase in cell migration and cell proliferation on right hand side resulting in the bending of primitive streak on the left-hand side. The vasculature develops on the right side, changing the circulation dynamics and forcing the other organs to the side of vasculature and hence the *situs inversus*. But the question remains as to why NO influences the cell movements only on the left side and not the right side. We delineate that NO driven BMP4 activation causes perturbation of CPC migration, resulting in the mis-orientation of the heart. Further work needs to be done to understand mechanistic insight of NO mediated embryonic rotation in relation to other major LR signalling pathways.

## 4. Materials and Methods

### 4.1. Chicken Embryos and Drug Treatment

White leghorn (WH) eggs were obtained from Poultry Research centre, Nandanam, Chennai. They were incubated at 38 °C in a humidified incubator as described previously [36], windowed and staged according to Hamburger and Hamilton (HH stage) [37]. The NO donors were dissolved in sterile water at mentioned concentrations. A small hole was made in the broad end of the day 0 egg with a sterile needle. 10 µL of NO donor, DEAN (100, 500 and 1000 µM in 1× PBS) or other NO donors (SNP, DETA) or quencher (cPTIO) were injected through the hole in the air sac using a sterile tip as a single dose at HH stages 0- HH stage 13 into the air sac. The eggs were then sealed with a sterile mediplast tape and incubated further from day 0 to day 12 depending on the experiment requirement. The control eggs were treated with 1X PBS or left without an injury in the egg shell. SNP and DEAN were obtained from Sigma-Aldrich, St. Louis while DETA, cPTIO were obtained from Cayman chemicals, Michigan. The authors declare that the research was conducted with prior approval from Institutional Biosafety and Ethical Committee (IBEC) of the AU-KBC Research Centre as per Annexure I; Project 02 (14-11-2011).

### 4.2. Zebrafish (Danio Rerio) Embryos and Drug Treatment

Zebrafish pairs (male and female) were bought from Ganesh Aqua firm, Chennai and maintained according to standard conditions (Siamwala et al., 2012). Fishes were bred at 28.5 °C on a 14 h/light and 10 h/dark cycle and allowed to mate naturally. Fertilized zebrafish eggs were collected from the bottom of the fish tank at the 1–2 cell stage, allowed to develop and staged according to hour post fertilization (hpf). Zebrafish embryos were grown in the PBS or 100 µM DEAN and/or 100 ng/mL of recombinant noggin from 1hpf up to 12hpf.

### 4.3. Real Time Imaging of Heart Looping

White leghorn (WH) eggs were treated with PBS or 500 µM DEAN as described earlier. Eggs from each group were opened at time points 30 h, 34 h, 38 h, 42 h 45 h, 47 h and 50 h of incubation to get the intermediate stages of cardiac looping. The albumin was removed and intact yolk was transferred into a (20 × 15 × 8 cm^3^) glass tray filled with distilled water (to avoid rupturing the vitelline membrane). Approximately one cm radius of membrane, surrounding the embryo, was cut using surgical scissor and transferred to 25 mm Petri dish containing 1× PBS. The membrane was washed 2–3 times carefully to remove the attached yolk without damaging the embryo. Bright field images of ventral side of the embryo were taken under inverted microscope (Olympus, Chennai, TN, India).

### 4.4. Free Radical Measurement in Cardiac Progenitor Cells

Cardiac progenitor cells were isolated from chick embryo at HH stage 10+ [38] and Immuno-fluorescence experiment was carried out as described previously to confirm CPC using CD34 marker. Cultured CPC were treated with indicated concentrations of DEAN and PBS and cPTIO. Following treatment, the level of H_2_O_2_, peroxinitrite and superoxide were measured using Amplex red, Dihydrorhodamine and Nitro Blue Tetrazolium respectively as per the protocol previously described [39].

### 4.5. Semi-Quantitative Reverse Transcriptase PCR

HH stage 4- HH stage 23 chick embryos (*n* = 3 pooled embryos) or individual organs such as brain, muscle and heart were isolated from 3–4 chick embryos using sterile procedures and powdered in liquid nitrogen. The embryos and the individual organs were washed thoroughly to remove yolk particles. Total RNA was isolated from PBS or DEAN or cPTIO treated whole embryos or organs using spin prep kit (Medox Inc, Chennai, TN, India). cDNA synthesis was performed on 200 ng of RNA using mulv reverse transcriptase (Finnzymes) and PCR was performed using 50 ng cDNA. For all genes analysed, 2 μL of the cDNA reaction was used for PCR amplification. Primers used for PCR amplification and internal probing, product size and the annealing temperatures are tabulated in Table 4. β-actin was used as an internal control. The PCR amplification of the cDNA remained linear after 30 cycles. Controls for DNA contamination in the RNA preparation were performed for all sets of primers with the identical procedure as for the RT-PCR but without reverse transcriptase. The PCR products were resolved on 1% agarose gel at 80 V using agarose gel electrophoresis system.

### 4.6. Immunohistochemistry

White leghorn (WH) eggs were treated with PBS or 500 µM DEAN as described earlier. Egg from each group was opened on 4th day and heart was dissected out from the embryo. The isolated hearts were fixed in 10% neutral buffered formalin. Then histological sections were prepared as described previously [40].

In brief, Formalin fixed heart section were cut vertically after embedding them in paraffin. Sections were deparaffinized & rehydrated sequentially. Then antigen retrieval was done using Tris-EDTA buffer (10 mM Tris base, 1 mM EDTA, 0.05% Tween 20, pH 9.0) at 95 °C for 20 min. Sections were washed with PBST. Nonspecific sites were blocked by 10% BSA. Then sections were immersed in polyclonal primary antibody diluted according to manufacturer (Abcam, Kolkata, WB, India) instructions (anti-BMP4 or anti-smad1 or anti-psmad1) in 0.5% BSA-Tween 20 and incubated overnight at 4 °C. In case of noggin treatment, sections were incubated with 100 ng/mL of recombinant noggin protein (Abcam) for 2 h at RT prior to addition of primary antibody. After washing with PBS sections were treated with 1:5000 dilution of FITC tagged anti-rabbit antibody and incubated for 30 min in dark at RT.

After washing with PBS, sections were imaged under fluorescence microscope in 32× magnification.

### 4.7. Western Blot Analyses

Embryos at the eight to 12 somite stages were removed from the eggs. Embryos were rinsed in cold 1× PBS and embryo or organ homogenates were prepared in 10 mL/embryo homogenization buffer. The embryos were then grinded to powder using by adding liquid nitrogen in a mortar and pestle. 100 µg proteins were subjected to 10% SDS/PAGE gel and transferred to nitrocellulose membranes (MDI 0.45 micron). Proteins were detected using rabbit polyclonal primary antibodies; anti-BMP4 (Calbiochem, 1:500; Bangalore, KN, India); anti-Noggin (Calbiochem, 1:500); anti-Notch (Calbiochem, 1:1000), anti-Shh (Abcam, 1:1000), anti-actin (Sigma, 1:1000; Bangalore, KN, India) overnight incubation, followed by incubation with ant-rabbit secondary antibodies for 2 h. The membrane was developed using TMB/H_2_O_2_ substrate.

### 4.8. Whole Mount Antibody Staining

Whole mount antibody staining was performed as described previously [41]. In brief, the HH stage 8 embryos were removed from the yolk and washed rigorously with 1 × PBS to remove the yolk completely. The embryos were then fixed with 4% paraformaldehyde. 0.3% H_2_O_2_ solution made in 0.5% TBST was added to inactivate the endogenous peroxidase followed by 3 half an hour washes. Blocking solution was prepared with 3% BSA in TBST and embryos incubated with blocking solution for an hour. Primary antibodies (1:1000) were added in the blocking solution and the embryos incubated at room temperature for 2 days. Next secondary antibodies (1:2000) were added and the embryos incubated for 1 day. The colour development was carried out using a DAB substrate and H_2_O_2_. Photographs were taken at 4× magnification using Nikon cool pix camera adapted to a stereomicroscope.

### 4.9. Cell Migration Assay Using Boyden Chamber

Cardiac progenitor cell migration under following treatment such as, 1 × PBS, DEAN, cPTIO and DM determined using Boyden chamber-based migration protocol as described previously [42]. Migrated cells were stained with DAPI (300 nM/mL) to count under 20× magnification with the help of a fluorescent microscope.

### 4.10. Statistical Analyses

All the experiments were performed in triplicates with *n* = 100 eggs or fishes unless mentioned otherwise. The data are represented as mean ± SEM of three independent experiments. The data were analysed using one-way ANOVA, two-tailed student’s *t*-tests, Mann-Whitney U and Tukey post hoc test as appropriate. *p* values ≤ 0.05 were considered to be significant.

## Figures and Tables

**Figure 1 ijms-20-01157-f001:**
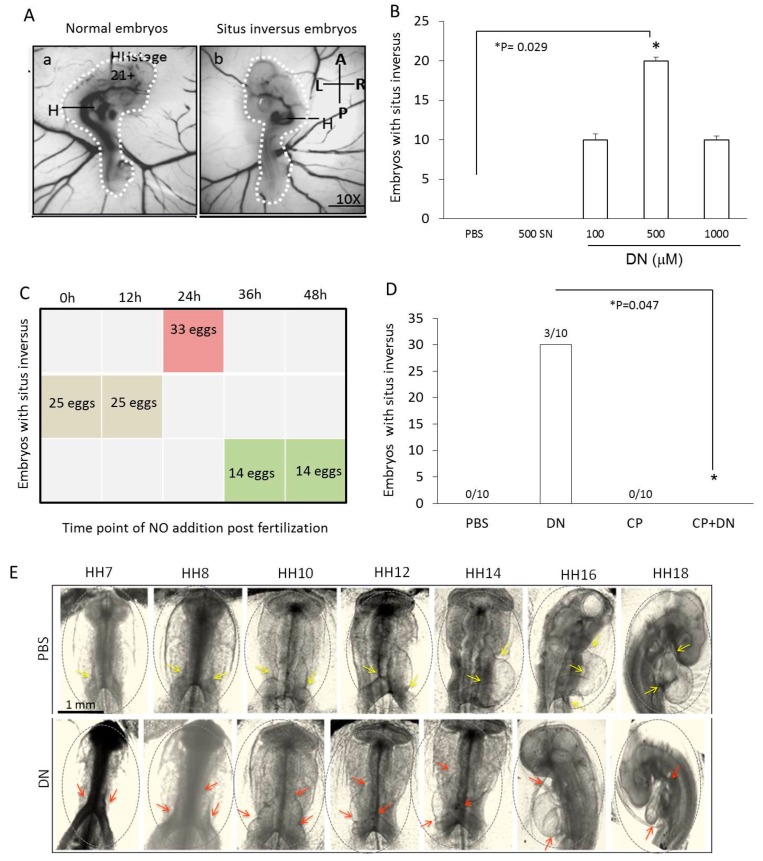
Dose and temporal effect of NO on heart orientation (*situs inversus*). (**A**, a–b) HH stage 7–8 eggs loaded with 1X PBS shows embryo on the normal side. DEAN (DN) loaded egg shows *situs inversus*. Dashed lines indicate the curvature of the embryo. Arrows show the position of the heart (H) in control and DN treated embryos *n* = 200 embryos. (**B**) 200 white leghorn eggs were loaded with PBS or sulphoNONOate (SN) or different DEAN concentrations (100 μM, 500 μM and 1000 μM) at HH stage 7–8. The results are expressed as mean +/−S.E. (*n* = 200, * *p* < 0.05 vs. PBS control). (**C**) Optimization of the time point for DN treatment to identify the most effective time for DN treatment in reference to number of embryos with *situs inversus*. (**D**) Endogenous NO effects were tested by loading the chick eggs at 24 h with DN or cPTIO (CP) or DN + CP and the number of surviving embryos with *situs inversus* were scored. (**E**) PBS and DN treated eggs were opened at HH7, HH8, HH10, HH12, HH14, HH16 and HH18. Real time imaging of heart looping images were taken under inverted microscope. PBS treated eggs showing the heart development and the curvature of the body on the right side. DEAN (DN) treated eggs showing the heart development and the curvature of the body on the left side. The grey circle indicates the head to heart region of the embryo. Yellow arrows indicate the curvature during the cardiac tube looping in PBS group (normal situs). Red arrows indicate the curvature during the cardiac tube looping in DN treated group (*situs inversus*).

**Figure 2 ijms-20-01157-f002:**
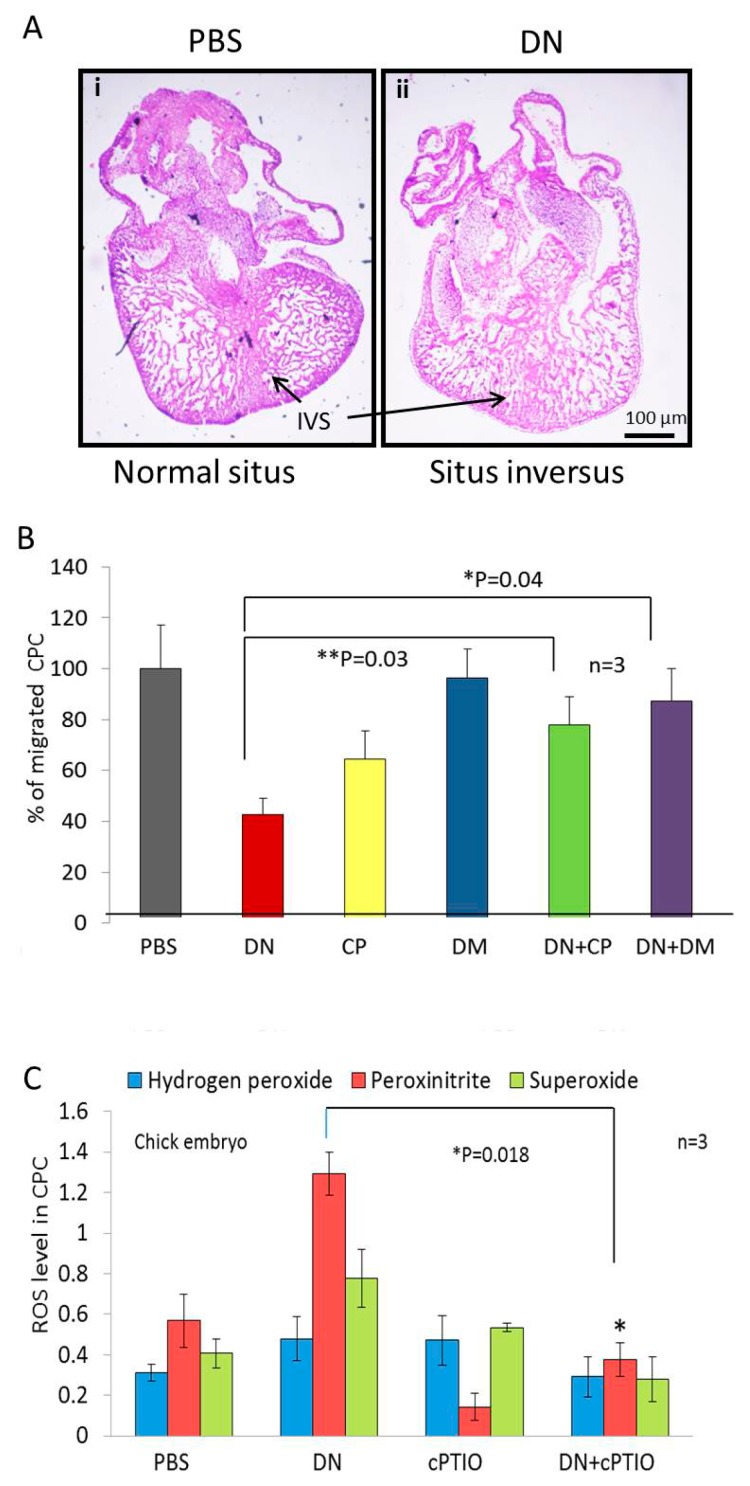
NO mediated migration of CPC migration and oxidative stress. (**A**) Vertical sections of 4th day old chick embryonic heart stained with H&E, arrows indicate intraventricular septum (IVS). (i) PBS treated embryonic heart section showing the prominent interventricular septum and ventricular polarity indicated by black arrows. (ii) Heart section of DN treated embryo showing loss of interventricular septum and ventricular polarity. (**B**) Boyden chamber analysis of CPC migration under PBS or DN or CP or Dorsomorphin (DM) or DN + CP or DN + DM. (*n* = 3, * *p* = 0.04 DN vs. DN + DM, ** *p* = 0.03 DN vs. DN + CP, by one-way ANOVA). (**C**) The cardiac progenitor cells (CPC) were isolated from the vascular bed and treated with DN or CP or DN + CP. Free radicals which are the main components of ROS machinery such as hydrogen peroxide, peroxynitrite and superoxide were measured from the CPC using Amplex red, dihydrorhodamine and nitro blue tetrazolium assays. (*n* = 10, * *p* < 0.05 vs. DN by one-way ANOVA, *n* = 3 embryos).

**Figure 3 ijms-20-01157-f003:**
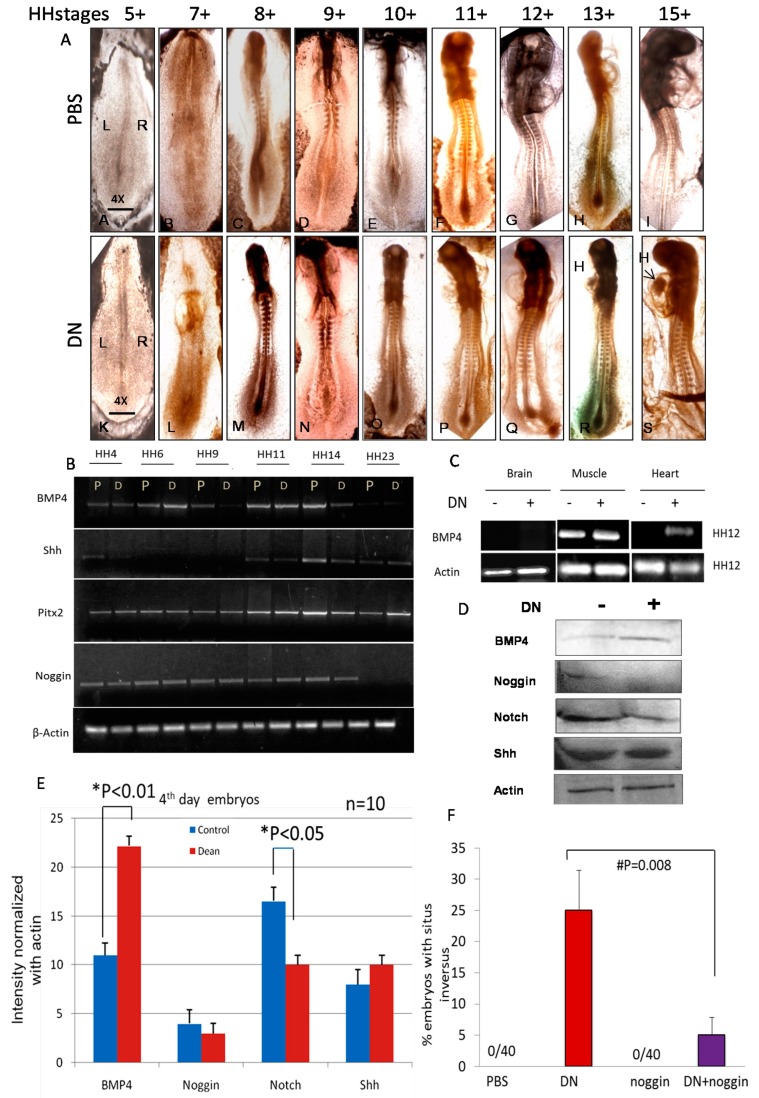
Expression profile of the proteins involved in the turning of the embryo. (**A**) Antisense BMP4 probes show the distribution of BMP4 gene expression in the PBS and DN treated sectioned heart lumen. (**B**) RT-PCR analysis of genes associated to BMP signalling (BMP4, Shh, Pitx2, Noggin) at different stages of embryonic development (HH4, HH6, HH9, HH11, HH14 and HH23 in PBS control and DN treated embryos. (**C**) BMP4 gene expression analysis in different organs (Brain, muscle and heart) at HH stage 12. (**D**) Western blot analysis of protein expression of genes associated with left right axis (BMP4, Noggin, Notch and Shh) in HHstage 8) embryo. (**E**) Densitometry results shown in parallel represent means ±SE of 3 independent experiments. * Significantly different compared with vehicle control. (**F**) Figure showing the number of zebrafish embryos with *situs inversus* under the treatment of DN followed by noggin, a selective inhibitor of BMP4. The results are expressed as mean ±S.E. (*n* = 40, # *p* < 0.008 DN vs DN + noggin by one-way ANOVA).

**Figure 4 ijms-20-01157-f004:**
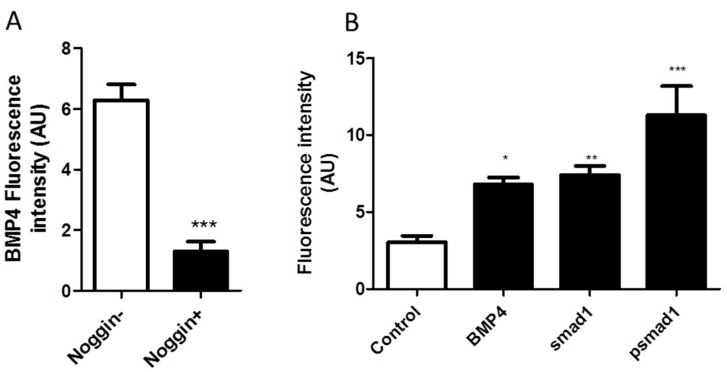
NO modulates BMP4-SMAD pathway in heart. (**A**) Specificity test of BMP4 antibody: Vertical sections of 4th day old chick embryonic heart were immune-stained with BMP4 antibody in two groups one pre-treated with BSA (indicated as noggin −) and another one with BSA+ Recombinant noggin (indicated as noggin +). Relative fluorescence intensity measurements using Adobe photoshop expressed as arbitrary units (AU). The results are expressed as mean +/− S.E. (*n* = 3, *** *p* < 0.001 BSA vs. BSA+noggin by *t*-test). (**B**) IHC of DN treated embryos showing higher expression of BMP4, SMAD and pSMAD in heart compared to that of PBS treated embryos. Relative fluorescence intensity measurements using Adobe photoshop expressed as arbitrary units (AU). The results are expressed as mean +/− S.E. (*n* = 3, * *p* < 0.05, ** *p* < 0.01, *** *p* < 0.001 PBS vs DN by one-way ANOVA).

**Figure 5 ijms-20-01157-f005:**
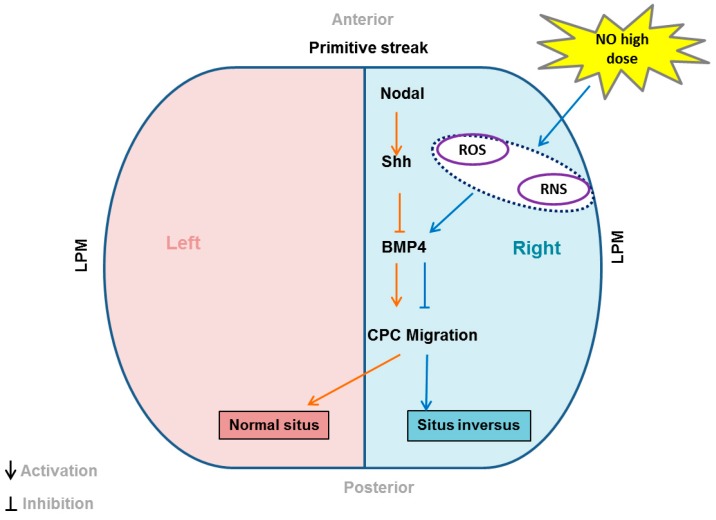
Schematic representation of the signalling events that occur in the presence of higher dose of NO resulting in *situs inversus*.

**Table 1 ijms-20-01157-t001:** Measurement of heart beats per minute using commercially available heart beat monitor.

	4th Day Avg. bpm	5th Day Avg. bpm	6th Day Avg. bpm
PBS	222	236	261
Normal	244	262	274
Situs inversus	265	280	297

**Table 2 ijms-20-01157-t002:** The organ positions of the chick and zebrafish embryos were scored and summarized in the table. ND, not determined; NA, not applicable.

Internal Organs	Heart Left (L), Right (R)		Liver		Lungs		Gut	
**PBS**	**YES**	**NO**	**NO**	YES	NO	YES	NO	YES
DEAN	144/200 embryos	56/200 embryos	NO	YES	NO	YES	NO	YES
PBS	YES	NO	NA	NA	ND	ND	ND	ND
DEAN	90/100 fishes	10/100 fishes	NA	NA	ND	ND	ND	ND

**Table 3 ijms-20-01157-t003:** Measurements of heart functions related parameters.

	Heart Contraction Area (Pixels)	Heart Relaxation Area (Pixels)	Vascular Bed Area (Pixels)
**PBS**	542.6	422.3	21.93
**SNP**	312	255	36.5
**DN100**	655	456	17.7
**DN500**	577	321	35.19
**DN1000**	677	492	15.88

**Table 4 ijms-20-01157-t004:** Semi-quantitative RT-PCR primers.

Gene of Interest	Product Size	Annealing Temperature	Primer Sequence
BMP4	911 bp	56.2 °C	Sense-5′ CTACTATGCCAAGTCCTGCT 3′ Anti-Sense-5′ TCGCTGAAATCCACATAGA 3′
Shh	508 bp	56.4 °C	Sense- 5′ GTAATTGGATTCAATGGTCG 3′ Anti-Sense-5′GGCCAGCATTCCGTACTT 3′
Pitx2	762 bp	61 °C	Sense-5′ATGAGTTGCATGAAGGACCC 3′ Anti-Sense-5′TGCTCACACGGGCCTGTCCA 3′
Nogg	672 bp	59 °C	Sense-5′ATGGATCATTCCCAGTGCCTTGT3′ Anti-Sense-5′CTAGCAGGAGCACTTGCACTC3′
Nodal	180 bp	60 °C	Sense-5′CTGGATCGTCTACCCCAAGA3′ Anti-Sense-5′ATGGAGAGGGGACTCATCCT3′
Wnt3a	485 bp	60 °C	Sense-5′GTTCTGCAGCGAAGTGGTG3′ Anti-Sense-5′GAGTGTCACAGCCGCAGATG3′
β-Actin	165 bp	60.5 °C	Sense- 5′ TCTGACTGACCGCGTTACTC 3′ Anti-Sense-5′ CCATCACACCCTGATGTCTG 3′

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
