# Peer review of "Nitric Oxide Reverses the Position of the Heart during Embryonic Development"

_ijms, 2019, doi:10.3390/ijms20051157_

Reviewer 1 Report

In the results section it is appropriate to comment on the experimental data obtained in the present study. On page 2 lines 76-77 it is appropriate to delete the sentence "Fujinaga (1990) was the first to show that nitrous oxide (N2O), caused laterality changes when70 timed pregnant rats were exposed to 75% N2O for 24h on day 8 of gestation [13]". It is also reported in the discussions section.

An extensive revision of the text is required. Figure 3, Table 1 and Table 3 show results related to experiments on zebrafish. However, the experimental part concerning zebrafish is not reported in the materials and methods. It would be better to add in materials and methods the part related to the experiments on zebrafish. Moreover it would be better to comment on obtained results both on chicken embryos and on zebrafish, justifing the choice of both experimental models.

Author Response

We thank the reviewer for the constructive criticism and time. We have revised the manuscript according to the reviewer’s comments.

Point 1:

In the results section it is appropriate to comment on the experimental data obtained in the present study. On page 2 lines 76-77 it is appropriate to delete the sentence "Fujinaga (1990) was the first to show that nitrous oxide (N2O), caused laterality changes when70 timed pregnant rats were exposed to 75% N2O for 24h on day 8 of gestation [13]". It is also reported in the discussions section.

Response 1: We have taken the reviewers suggestion into consideration and removed the sentence "Fujinaga (1990) was the first to show that nitrous oxide (N2O), caused laterality changes when 70 timed pregnant rats were exposed to 75% N2O for 24h on day 8 of gestation [13]". The text has been revised accordingly in the results and discussion sections which can be found at page 2 line 77 and page 12 line 239  respectively.

Point 2:

An extensive revision of the text is required. Figure 3, Table 1 and Table 3 show results related to experiments on zebrafish. However, the experimental part concerning zebrafish is not reported in the materials and methods. It would be better to add in materials and methods the part related to the experiments on zebrafish. Moreover it would be better to comment on obtained results both on chicken embryos and on zebrafish, justifing the choice of both experimental models.

Response 2: We thank the reviewer for pointing the missing section. The text has been revised and an additional section in methods has been added at page 9 “4.2 Zebrafish (Danio rerio) embryos and drug treatment Zebrafish pairs (male and female) were bought from Ganesh Aqua firm, Chennai and maintained according to standard conditions (Siamwala et al., 2012). Fishes were bred at 28.5°C on a 14h/ light/ 10h dark cycle and allowed to mate naturally. Fertilized zebrafish eggs were collected from the bottom of the fish tank at the 1-2 cell stage, allowed to develop and staged according to hour post fertilization (hpf). Zebrafish embryos were grown in the PBS or 100µM DEAN and/or 100ng/ml of recombinant noggin from 1hpf upto 12hpf.”

Table 3  (page 6 line 146) has been removed.

Statement for the choice of both experimental models has been added at page 12 lines 250-252 as “Preliminary studies showed that frequency of SI was higher in chick embryos (28%) than in zebrafish (10%) (table 1) under DEAN treatment. Therefore, further studies were performed on chick embryo model.”

Reviewer 2 Report

Nitric oxide (NO) is an essential signal molecule to maintain cellular homeostasis. Behind these unusual capabilities of NO is the chemistry of this molecule, an unstable, reactive, free radical and short half-life gas. It is a lipophilic molecule that crosses all the barriers that biological membranes can impose. A family of NO synthases (NOS) catalyzes the reaction. We can distinct four genetically different types of NOS: endothelial (eNOS), neuronal (nNOS), inducible (iNOS) and mitochondrial (mtNOS). NO produced by eNOS plays crucial roles in cardiac homeostasis. The objective of the present study was to determine the global effects of NO overexpression in developing cardiovascular system. The Authors carried out experiments with chicken embryos. The chick embryo is an excellent model for left-right determination studies, since the embryos can be manipulated as early as the blastula stage, independent of the maternal effects. In addition, to confirm NO specific effects and to rule out the non-specific effects of DEAN (a NO donor), the Authors used NO donors with different half-lives tested. In the present study it was shown that high levels of NO switches the heart position from the right to the left resulting in situs inversus. NO reverses the cardiac development by altering the cardiac progenitor cell migration. Overall this is a well-designed study and provides very interesting results.

I have found only some minor mistakes:

Page 2, line 51: “…and phosphodiesterase inhibitors (PDE) ameriolate erectile dysfunction.” I suggest replacing this statement with the information about the use of PDE inhibitors e.g. sildenafil in the therapy of neonatal cardiovascular disorders. It will be more connected with the subject.

Some corrections:

-          The abbreviations should be used consequently eg. NO (page 2, line 44, line 71)

-          The abbreviation of BMP4 is appeared for the first time in the introduction thus it should be explained there (page 2, line 63)

-          Similarly with the abbreviation of HH (page 2, line 86)

Author Response

We thank the reviewer for the constructive criticism and time. We have revised the manuscript according to the reviewer’s comments.

Point 1:

Nitric oxide (NO) is an essential signal molecule to maintain cellular homeostasis. Behind these unusual capabilities of NO is the chemistry of this molecule, an unstable, reactive, free radical and short half-life gas. It is a lipophilic molecule that crosses all the barriers that biological membranes can impose. A family of NO synthases (NOS) catalyzes the reaction. We can distinct four genetically different types of NOS: endothelial (eNOS), neuronal (nNOS), inducible (iNOS) and mitochondrial (mtNOS). NO produced by eNOS plays crucial roles in cardiac homeostasis. The objective of the present study was to determine the global effects of NO overexpression in developing cardiovascular system. The Authors carried out experiments with chicken embryos. The chick embryo is an excellent model for left-right determination studies, since the embryos can be manipulated as early as the blastula stage, independent of the maternal effects. In addition, to confirm NO specific effects and to rule out the non-specific effects of DEAN (a NO donor), the Authors used NO donors with different half-lives tested. In the present study it was shown that high levels of NO switches the heart position from the right to the left resulting in situs inversus. NO reverses the cardiac development by altering the cardiac progenitor cell migration. Overall this is a well-designed study and provides very interesting results.

Response 1: We thank the reviewer for extensive reading, understanding and appreciating our findings.

Point 2:

Page 2, line 51: “…and phosphodiesterase inhibitors (PDE) ameriolate erectile dysfunction.” I suggest replacing this statement with the information about the use of PDE inhibitors e.g. sildenafil in the therapy of neonatal cardiovascular disorders. It will be more connected with the subject.

Response 2: We thank the reviewer for suggestion. The sentence has been changed from “Inhaled NO is used for the treatment of pulmonary hypertension in new born, pre-term neonates with hypoxic respiratory failure [11] and phosphodiesterase inhibitors (PDE) ameriolate erectile dysfunction” to “Inhaled NO is used for the treatment of pulmonary hypertension in new born, pre-term neonates with hypoxic respiratory failure [11] and phosphodiesterase inhibitors (PDE) e.g. sildenafil therapy to improve the hemodynamics in children with pulmonary hypertension [12].” With new reference at Page 2, line 51.

Point 3:

The abbreviations should be used consequently eg. NO (page 2, line 44, line 71)

Response 3: The text has been checked for abbreviations and revised with track changes at respective places (page 2, line 44, line 74).

Point 4:

The abbreviation of BMP4 is appeared for the first time in the introduction thus it should be explained there (page 2, line 63)

Response 4: The text has been revised and highlighted with track change at respective places page 2, line 64.

Point 5:

Similarly with the abbreviation of HH (page 2, line 86)

Response 5: The text has been revised and highlighted with track change at respective places (page 2, line 89).

Round  2

Reviewer 1 Report

Accept in the present form